# Genistein Activates Transcription Factor EB and Corrects Niemann–Pick C Phenotype

**DOI:** 10.3390/ijms22084220

**Published:** 2021-04-19

**Authors:** Graciela Argüello, Elisa Balboa, Pablo J. Tapia, Juan Castro, María José Yañez, Pamela Mattar, Rodrigo Pulgar, Silvana Zanlungo

**Affiliations:** 1Departament of Gastroenterology, School of Medicine, Pontificia Universidad Católica de Chile, Santiago 3580000, Chile; ebalboa@gmail.com (E.B.); pablo.tapia.o@gmail.com (P.J.T.); jcastros@puc.cl (J.C.); myanezh@gmail.com (M.J.Y.); 2School of Health Sciences, Universidad Católica del Maule, Talca 3466706, Chile; 3Center for Cell Biology and Biomedicine (CEBICEM), School of Science and Medicine, Universidad San Sebastián, Santiago 7510235, Chile; 4Departament of Physiology, School of Biological Science, Pontificia Universidad Católica de Chile, Santiago 3580000, Chile; pamela.mattararanguiz@einsteinmed.org; 5Laboratory of Genomics and Genetics of Biological Interactions, Instituto de Nutrición y Tecnología de los Alimentos (INTA), Universidad de Chile, Santiago 7830490, Chile; rpulgar@inta.uchile.cl

**Keywords:** Niemann–Pick C, TFEB, genistein, lysosomal storage diseases, cholesterol, lysosome clearance

## Abstract

Niemann–Pick type C disease (NPCD) is a lysosomal storage disease (LSD) characterized by abnormal cholesterol accumulation in lysosomes, impaired autophagy flux, and lysosomal dysfunction. The activation of transcription factor EB (TFEB), a master lysosomal function regulator, reduces the accumulation of lysosomal substrates in LSDs where the degradative capacity of the cells is compromised. Genistein can pass the blood–brain barrier and activate TFEB. Hence, we investigated the effect of TFEB activation by genistein toward correcting the NPC phenotype. We show that genistein promotes TFEB translocation to the nucleus in HeLa TFEB-GFP, Huh7, and SHSY-5Y cells treated with U18666A and NPC1 patient fibroblasts. Genistein treatment improved lysosomal protein expression and autophagic flux, decreasing p62 levels and increasing those of the LC3-II in NPC1 patient fibroblasts. Genistein induced an increase in β-hexosaminidase activity in the culture media of NPC1 patient fibroblasts, suggesting an increase in lysosomal exocytosis, which correlated with a decrease in cholesterol accumulation after filipin staining, including cells treated with U18666A and NPC1 patient fibroblasts. These results support that genistein-mediated TFEB activation corrects pathological phenotypes in NPC models and substantiates the need for further studies on this isoflavonoid as a potential therapeutic agent to treat NPCD and other LSDs with neurological compromise.

## 1. Introduction

Niemann–Pick type C disease (NPCD) is a fatal lysosomal storage disease (LSD) caused by mutations in *NPC1* or *NPC2* genes, whose protein products cooperatively mediate the efflux of cholesterol from late endosomes and lysosomes [1]. Loss of function in NPC1 or NPC2 proteins leads to the progressive accumulation of cholesterol in lysosomes, causing lysosomal dysfunction [2], oxidative stress [3], autophagy defects [4], mitochondrial dysfunction [5], and cell death [6]. The most affected tissues are the brain, liver, and lungs, leading to death in NPC patients due to hepatic failure or neurodegeneration [7,8].

Lysosomes have been identified as an essential part of cell metabolism, influencing processes as diverse as nutrient sensing, secretion, gene regulation, plasma membrane repair, and cholesterol transport [9]. Several studies have reported that, in addition to lysosomal cholesterol accumulation, defective autophagic flux is involved in the pathogenesis of NPCD [10,11,12,13]. It has been reported that cholesterol load reduces lysosomal enzymatic activity [14], inhibits endosome–lysosome trafficking [15], lowers the ability of lysosomes to fuse with endocytic and autophagic vesicles [16,17], and blocks autophagic flux in the autophagy pathway, supporting the idea of defective autophagic flux involvement [18]. Moreover, recently an alteration in gene expression of autophagy-associated genes in NPC skin fibroblasts from patients has been found [19].

Autophagy is a highly conserved biodegradation process that is involved in lipid metabolism [20] and is also responsible for important cellular processes, such as the clearance of long-lived proteins and damaged organelles. Therefore, resetting autophagy and reducing cholesterol accumulation in the lysosome are necessary to treat NPCD. On the other hand, transcription factor EB (TFEB) is the master regulator of the coordinated lysosomal expression and regulation (CLEAR) network, including genes related to lysosomal biogenesis, autophagy, exocytosis, and lysosomal function [9,21,22,23,24]. TFEB subcellular localization and activity are regulated by mechanistic mTORC1-mediated phosphorylation, which occurs on the lysosomal surface. Phosphorylated TFEB is retained in the cytoplasm, whereas dephosphorylated TFEB translocates to the nucleus to induce the transcription of the CLEAR network [9,21,22,25].

Notably, TFEB overexpression or activation induces the degradation of bulk autophagy substrates, such as long-lived proteins [26], and clears out lipid droplets [27], indicating that TFEB plays a role in modulating organelle-specific autophagy, such as lipophagy. TFEB also induces lysosomal exocytosis [24,28,29], a process by which lysosomes fuse to the plasma membrane and secrete their content to the extracellular space. This is relevant in diseases where the cells’ degradative capacity is compromised. Indeed, several in vitro and in vivo studies suggest that lysosomal exocytosis induction could decrease the accumulation of storage materials in lipid storage disorders [30,31,32]. In fact, it has been reported that TFEB overexpression promotes cellular clearance, hence ameliorating phenotypes in several diseases, such as multiple sulfatase deficiency [24], Pompe disease [33,34], alpha1-antitrypsin deficiency [35], as well as Alzheimer’s, Parkinson’s, and Huntington’s disease, among other neurodegenerative diseases [36,37,38]. Indeed, multiple studies have shown that the exogenous expression of TFEB and the pharmacological activation of endogenous TFEB attenuate disease phenotypes in animal models of LSDs [24,33,39,40]. This has led to the proposal of TFEB activation as a therapeutic target for LSD treatment. Our group has recently shown that TFEB activation through c-Abl inhibition promotes cellular clearance in NPC models [41].

Among natural modulators of TFEB, genistein seems promising, given that this isoflavonoid can cross the brain–blood barrier (BBB) [42], which is necessary for the treatment of neurological symptoms present in most LSD included NPCD. Along this line, it has been shown that genistein decreases urinary and tissue glycosaminoglycans (GAG) levels, including GAG deposits in the brains of mucopolysaccharidoses (MPS) II mice after 10 weeks of treatment with a dose of 25 mg/kg/day [43]. In another study, MPS IIIB mice treated with a high dose of genistein (160 mg/kg/day) exhibited a significant decrease in heparan sulfate accumulation and neuroinflammation in the brain, as well as improvement in the behavioral phenotype [44], confirming that genistein can reach the brain to exert its effect. Moreover, treating MPSIII patients with elevated doses of a genistein-rich soy extract improves neurological symptoms, such as their ability to walk, reactions to surrounding stimuli, communication skills, and sleep [45]. Such beneficial effects may be associated with the inhibition of TFEB-induced glycosaminoglycan synthesis previously described in genistein-treated MPSII cells [46].

On the other hand, Pierzynowska et al. proposed genistein in treating Huntington’s disease, given the results seen from cell lines treated with genistein, where levels of mutated Huntington and number of aggregates significantly decreased, and where autophagy increased, leading to greater cell viability [47]. Along similar lines, it has been reported that TFEB activation by genistein promotes reductions in cystine storage in cystinotic cells and rescues lysosomal abnormalities in LSD nephropathic cystinosis [48]. Together, these findings further suggest that stimulating TFEB activity through genistein may represent a useful therapeutic tool to decrease the pathogenic features of NPC cells, including lysosomal functions and autophagy alterations, as well as the pathological accumulation of cholesterol.

In this study, we investigated the effect of endogenous TFEB activation with genistein on the NPC phenotype in different cellular models. Our data show that genistein promotes TFEB translocation to the nucleus in all the NPC models used, including pharmacological models of HeLa TFEB-GFP, Huh7, and SHSY-5Y treated with the U18666A (U18) drug, as well as NPC1 patient fibroblasts as genetic models. We observed improved lysosomal protein expression and autophagic flux in NPC1 patient fibroblasts treated with genistein. These results were associated with reductions in p62 levels and higher LC3-II levels. Moreover, genistein induced greater β-hexosaminidase activity in culture media of NPC1 patient fibroblasts, suggesting an increase in lysosomal exocytosis. This result correlated with a reduction in cholesterol accumulation after filipin staining in all NPC models used, including pharmacological and genetic models.

## 2. Results

### 2.1. Genistein Promotes TFEB Nuclear Translocation in NPC Pharmacological and Genetic Models

TFEB activity is regulated according to its subcellular localization. Inactive TFEB is retained in the cytoplasm, whereas activated TFEB is translocated to the nucleus to induce the transcription of its target genes. To determine if genistein activates TFEB in NPC models, we evaluated TFEB nuclear localization. First, we treated HeLa TFEB-GFP cells with 0.5 µg/mL of hydrophobic polyamine U18, a cell-permeable small-molecule inhibitor of the endosomal cholesterol transporter NPC1 [49], to induce the NPC phenotype. Then we applied different amounts of genistein (50 µM, 100 µM, and 150 µM) for 24 h. Subsequently, we measured TFEB-GFP nuclear localization using a high-content nuclear translocation assay in a confocal microscope. We used 0.3 µM Torin 1, an mTORC1 inhibitor, for 3 h as a positive control for TFEB nuclear translocation. As expected, Torin 1 increased the nuclear localization of TFEB (Figure 1A,B). U18 promoted a significant increase in the TFEB-GFP nuclear signal compared to control conditions. Furthermore, a significant increase in the TFEB-GFP nuclear signal was observed at 50 µM, 100 µM, and 150 µM of genistein application, being the last concentration the most effective in inducing TFEB translocation (Figure 1A,B). Although U18 0.5 alone increased the TFEB-GFP nuclear signal compared to the control condition, co-treatment with genistein promoted an even higher and significant increase in the TFEB-GFP nuclear signal. To quantify TFEB intracellular distribution, we performed nuclear-cytoplasmic fractionation experiments. Immunoblotting analysis of the cytoplasmic and nuclear fractions showed that treatment with genistein significantly promoted TFEB nuclear translocation in HeLa TFEB-GFP cells treated with U18 (Figure 1C,D), confirming our previous result when using the high-content nuclear translocation assay.

In addition, and considering that some of the most affected tissues in NPCD are the liver and the brain, we tested genistein in cellular models of human hepatic Huh7 (Figure 1E,F) and the neuroblastoma SHSY-5Y cell line (Figure 1G,H). Both were treated with U18 and genistein for 24 h. Similar to HeLa cells, immunoblotting analysis of the cytoplasmic and nuclear fractions showed that U18 caused a significant increase in the TFEB nuclear signal in both NPC models compared to control conditions and that co-treatment with genistein significantly increased endogenous nuclear TFEB levels (Figure 1F–H).

We then decided to analyze the effects of genistein in GM05659 WT fibroblast and GM03123 NPC1 patient fibroblasts carrying a missense mutation 709 (C > T) in exon 6 of the *NPC1* gene, which resulted in the substitution of a serine for a proline at codon 237. The second allele also carries a missense mutation T > C at nucleotide 3182 (3182T > C) in exon 21, which results in the substitution of a threonine for an isoleucine at codon 1061 in a transmembrane domain. Immunoblotting analysis of the cytoplasmic and nuclear fractions showed that the nuclear–cytoplasmic ratio of TFEB protein increased in NPC1 patient fibroblasts compared to wild type (WT) fibroblasts in the basal condition. However, treatment with genistein significantly increased the nuclear–cytoplasmic ratio of TFEB protein compared to its basal conditions in both WT and NPC1 patient fibroblasts (Figure 1I,J). These results, when taken together, indicate that genistein can induce TFEB nuclear translocation in different NPC models.

### 2.2. Genistein Induces TFEB Target Gene Transcription in NPC1 Patient Fibroblasts

Previous studies have shown that translocation of TFEB to the nucleus correlates with an increase in the expression of CLEAR network genes [21,22,25]. To determine whether enhanced nuclear localization correlated with the upregulation of CLEAR genes, we analyzed mRNA levels of six direct TFEB target genes in WT and NPC1 patient fibroblasts treated with genistein at 150 µM for 24 h with real-time qPCR (Figure 2A). In response to genistein in WT and NPC1 patient fibroblasts, real-time qPCR analyses confirmed significant increases in mRNA levels of TFEB and most TFEB target genes, such as *LAMP1* (lysosomal membranes), *CTSB* and *CTSD* (lysosome hydrolases), *HATP6V1H* (ATPase responsible for acidifying intracellular compartments), and *TPP1* (a lysosomal serine protease with endopeptidase activity, shown to cleave peptides between hydrophobic residues) (Figure 2A). The upregulation of TFEB mRNA levels correlated with a significant increase in TFEB protein levels in response to genistein in NPC1 patient fibroblasts at 3, 6, 12, and 24 h of treatment (Figure 2B,C). These results indicate that genistein increases TFEB expression levels in NPC cells in addition to translocating it to the nucleus and subsequently activating it. Furthermore, TFEB upregulation was observed to induce its own transcriptional activation through an autoregulatory feedback loop, exerting global transcriptional control in response to different metabolic signals [27], which in this case was probably due to genistein signaling.

### 2.3. Genistein Induces Autophagy in NPC1 Patient Fibroblasts

Several studies have shown that TFEB controls the expression of genes involved in different steps of the autophagy process [22,26] and, as a result, TFEB activation induces a sharp increase in autophagy flux [50]. To determine whether enhanced TFEB nuclear localization is correlated with the upregulation of autophagy-related genes, we used real-time qPCR to assay the expression profile of a select set of previously identified TFEB target genes [26], such as *BECN1* (autophagy induction), *ATG9B* and *WIPI1* (early stages of autophagosome formation), and *VPS11 (*autophagosome/lysosome fusion). In addition, we analyzed mRNA levels of *MCOLN1*, also known as TRPML1 (transient receptor potential cation channel, mucolipin subfamily, member 1), the main Ca^2+^ release channel from the lysosomal lumen to the cytosol. MCOLN1 activity is relevant to several processes, including autophagosome-lysosome fusion and lysosomal exocytosis [24]. TFEB regulates lysosomal exocytosis by transcriptionally activating tethering factors and proteins involved in lysosomal dynamics, docking and fusion with the plasma membrane [51], doing so by first promoting lysosome recruitment and docking to the plasma membrane, after which Ca^2+^-mediated fusion occurs by means of MCOLN1 channel activation [28].

Real-time qPCR analyses showed significant increases in mRNA levels for all genes analyzed in response to genistein in WT and NPC1 patient fibroblasts (Figure 3A). Higher levels of mRNA in genes involved in initiating autophagy, as well as in autophagosome/lysosome fusion and lysosomal exocytosis, indicate that TFEB activation by genistein treatment regulates the transcriptional activation of autophagy in NPC1 patient fibroblasts, also suggesting that autophagy flux is induced in the NPCD genetic model.

It should also be noted that impaired autophagy is a pathological hallmark in NPC cells [10]. Thus, to test genistein effects on autophagy in NPC cells, we treated NPC1 patient fibroblasts with genistein at 150 µM for 1, 3, 6, 12, and 24 h (data not shown), and observed differences compared to control condition, only at 1 and 3 h. As a positive control for autophagy inhibition and autophagy activation, we used chloroquine and starvation media (Stv), respectively, for 3 h. We evaluated genistein effects upon autophagy in NPC1 patient fibroblasts by measuring the classic autophagy markers LC3 (I and II) and p62, also called sequestosome 1 (SQSTM1). LC3I is a cytosolic protein, and LC3II is bound to autophagosome membranes, where it eventually degrades into autolysosomes by lysosomal hydrolases when autophagy is induced [52]. However, LC3 II levels can also increase due to higher numbers of autophagosomes when autophagy is induced. Therefore, it is necessary to complement these results with other autophagy markers, such as p62, a protein that is incorporated into the autophagosome and degraded by autophagy, serving as a marker to study autophagic flux [53,54]. When autophagy is inhibited, p62 accumulates. However, when autophagy is induced, p62 decreases.

We observed that LC3 II levels were similar between Stv and genistein treatment at 1 and 3 h, and genistein significantly increased LC3 II levels at 1 h of treatment compared to the control condition (Figure 3B,C). On the other hand, we observed similar levels of p62 between the control condition and CQ treatment, which suggests that autophagy was altered in NPC1 patient fibroblasts. We also observed a downward trend in p62 levels at 1 and 3 h of genistein treatment and Stv media compared to the control condition. However, the observed difference was not statistically significant (Figure 3B–D). Additionally, immunofluorescence analysis showed that the NPC1 patient fibroblasts had higher levels of p62 compared to WT, confirming a decrease in p62 signals for NPC1 patient fibroblasts treated with genistein for 3 h compared to untreated fibroblasts (Figure 3E,F). Altogether, these results suggest that genistein treatment has a rapid effect (between 1 and 3 h) on autophagy activation in NPC1 patient fibroblasts.

### 2.4. Genistein Induces Lysosomal Exocytosis and Alleviates Cholesterol Accumulation in NPC Cells

It is well-known that TFEB activation induces exocytosis and lysosomal clearance [22,24]. In order to evaluate whether genistein could induce lysosomal exocytosis, we measured the activity of the lysosomal enzyme β-Hexosaminidase in culture media of NPC1 patient fibroblasts after genistein treatment for 1, 3, 6, 8, 12, and 24 h (Figure 4A). We observed a significant increase in β-Hexosaminidase activity in the culture media of NPC1 patient fibroblasts treated with genistein at 1 and 24 h compared to untreated fibroblasts. These results suggest that exocytosis is induced due to genistein treatment in NPC1 patient fibroblasts. Accordingly, given that one of the biochemical hallmarks of NPCD is the accumulation of free cholesterol in vesicles of the endosomal/lysosomal system, we explored whether the effect of genistein on autophagic flux and exocytosis observed in NPC1 patient fibroblasts resulted in a decrease in cholesterol accumulation. To this end, we measured cholesterol levels using filipin staining, a fluorescent polyene that specifically binds to free cholesterol [55] in NPC models for 24 h.

As expected, NPC models Huh7 and SHSY-5Y treated with U18, as well as NPC1 patient fibroblasts, showed higher filipin positive signals compared to controls or WT fibroblasts (Figure 4B–E). It is worth noting that genistein significantly lowered the filipin positive signal in all the NPC models used (Figure 4B–E). However, the effect of genistein was more significant in the acute NPC models (Huh7 and SHSY-5Y cells treated with U18). These results, taken together, suggest that genistein treatment induces lysosomal cholesterol release through autophagy and lysosome exocytosis inductions by TFEB activation in NPC models.

## 3. Discussion

Our results support that genistein controls cellular clearance by regulating TFEB activity in different NPCD models. In addition, they are in agreement with the idea that genistein can stimulate endogenous TFEB activity [56] and that TFEB activation can promote cellular clearance [57]. These results provide a promising therapeutic alternative for treating disorders that involve the accumulation of undegraded substrates into lysosomes. Hence, our aim was to investigate the effects of TFEB activation on NPCD, which is a fatal neurodegenerative disease characterized by the pathogenic accumulation of cholesterol in lysosomes. Our results support that genistein activates TFEB and corrects the NPC phenotype by inducing autophagy flux and lysosomal exocytosis as well as by reducing lysosomal cholesterol accumulation.

In this study, we used pharmacological and genetic models to visualize genistein effects on both acute and chronic NPCD. We observed that TFEB localization was predominantly nuclear in the different NPC models. However, genistein triggered an increase in TFEB nuclear localization. Genistein had a greater effect on the pharmacological NPC model induced with U18, perhaps because this is an acute model of the disease, where cholesterol accumulation is induced for some hours using the U18 drug, compared to the chronic model of accumulation, such as NPC1 patient fibroblasts. It is also important to consider that NPC is a chronic progressive disease, which suggests the need for early treatment in order to achieve better effects. Greater TFEB nuclear localization in NPC cells could also be interpreted as an attempt to compensate for lysosomal stress caused by lysosomal dysfunction and lower autophagy flux in NPC cells, as recently proposed [9,41]. In this context, physiological TFEB induction does not seem to counteract disease progression fully, and greater TFEB nuclear translocation by genistein was effective in decreasing lysosomal abnormalities in NPC models. We propose that this effect is most likely to be the consequence of TFEB’s ability to regulate lysosome function through the concomitant induction of lysosomal autophagy and exocytosis. In fact, we observed in NPC fibroblast that genistein was capable of inducing autophagy-related genes, such as *ATG9B*, *BECN1*, *MCOLN1*, *VPS11,* and *WIPI1*. Recently, the increase in genes related to the inhibition of autophagy and the decrease in those related to the induction of autophagy have been reported for NPC1 patient fibroblasts [19]. In the future, it would be interesting to evaluate whether genistein is also able to regulate the expression of these genes. Autophagy induction by genistein, which is also reflected by lower p62 signals and greater LC3-II, is an important finding, given that autophagy dysfunction has been correlated with cellular toxicity in NPCD [18].

Although the effects of genistein on TFEB nuclear translocation were observed mainly at 24 h of treatment, an increase in the activity of the ß-Hexosaminidase enzyme in the medium was observed at 1 and 24 h. It is possible to speculate that the early effect of genistein on β-Hexosaminidase activity can be due to another target of action rather than exocytosis, because a reduction in lysosome cholesterol accumulation associated with exocytosis activation was observed at 24 h of genistein treatment.

It is important to consider that treating NPC models with genistein can be deemed a healthier strategy than directly inhibiting mTOR, given that this kinase fulfills multiple functions [58]. However, although the induction of TFEB activity seems promising in treating several diseases, the side effects of its long-term activation must be considered.

It should be highlighted that treatment with 2-hydroxypropyl-β-cyclodextrin (HPβCD) in the NPC1 murine model, an auspicious experimental therapy for NPC disease, has shown to increase sterol flow into the cytoplasm through independent mechanisms by NPC1 and NPC2 proteins [59]. Along the same line, a recent report revealed that HPβCD alleviates cholesterol storage in NPC cells by inducing lysosomal exocytosis [60]. However, the main limitations of HPβCD are its toxicity and poor penetration across the BBB. Genistein is a nontoxic FDA-approved drug that can cross the BBB. Our results indicate that it activates lysosomal exocytosis (greater MCOLN1 mRNA abundance and β-Hexosaminidase activity) in NPC1 patient fibroblasts. This evidence gives genistein the especially desirable potential to treat NPCD, other LSDs, and neurodegenerative diseases involving lysosomal dysfunction, such as Alzheimer’s and Parkinson’s [61]. Some studies have indicated that genistein could be considered for genetic diseases that are still untreatable, including Cystic Fibrosis (CF), Huntington’s disease, and MPS [47,62,63,64]. Genistein is actually one of the most efficient activators of the CFTR mutant protein that causes CF diseases, considering that it can enhance expression, proper localization, and activity of the mutant protein [65,66]. Likewise, a pilot clinical study with ten patients suffering from the LSDs MPS IIIA and MPS IIIB indicated that a genistein-rich soy isoflavone extract administered by mouth daily for 12 months at 5 mg of genistein per 1 kg of body weight resulted in statistically significant improvements on all parameters tested. Such improvements included lower urinary glycosaminoglycan levels and higher psychological test scores measuring patients’ cognitive functions. Importantly, no significant side effects were observed during the clinical study, which indicated that this treatment is safe [67]. In addition, Kim et al. showed that therapy involving a high dose of genistein aglycone (150 mg/kg/day, for 12 months) is safe for patients with MPS involving the central nervous system [44]. However, the beneficial effects of genistein are controversial. Kingma et al. reported on increased levels of GAG in MPS I fibroblast treated with genistein [68] and that MPS I mice fed with a diet supplemented with genistein (corresponding to a dose of approximately 160 mg/kg/day) for 8 weeks presented adverse effects, such as decreased body length as well as the length of the femur, scrotal hernia, and/or scrotal hydrocele [69]. On the other hand, studies in MPS III patients supplemented with 5 mg/kg/day and 10 mg/kg/day of genistein for 1 year indicated that these doses were insufficient to improve disability and show clinical efficacy, respectively [70,71]. These results suggest the need for caution in using genistein, at least in patients with MPS I, and show the need for more studies to elucidate the pathways that are affected by genistein and which alter GAG metabolism so as to evaluate the therapeutic potential of genistein for MPS patients.

Our results showed that genistein can bypass NPC1 deficiency and decrease lysosomal abnormalities through TFEB activation in NPC models. However, we cannot exclude the direct action of genistein on the mutant protein NPC1 in NPC1 fibroblasts, which could enhance its residual activity as has previously been described for other mutant proteins in genetic diseases, such as CF and MPS.

Within the contexts of different cellular and mouse disease models, where autophagic/lysosomal substrates accumulate, several studies have shown that the induction of lysosomal biogenesis and autophagy through viral-mediated overexpression of TFEB can lead to the intracellular clearance of accumulating substrates as well as the significant correction of the disease phenotype [9,24,72]. In line with this, it would be of interest to analyze the effects of genistein on progressive neurodegeneration in NPC animal models and appropriate clinical trials so as to follow its effects on central nervous system abnormalities emerging from this disease.

Finally, our results position genistein as an interesting FDA-approved drug for the treatment of diseases in which lysosomes and TFEB have been proposed as therapeutic targets, such as the LSDs NPC, Batten, Gaucher, MPS, and Pompe disease, among others [36,57] as well as diseases with lysosomal dysfunction and neurological compromises, such as Alzheimer’s and Parkinson’s [61].

In summary, this study supports that genistein-mediated TFEB activation corrects pathological phenotypes in NPC models and substantiates the need for further studies of this isoflavonoid as a potential therapeutic agent to treat NPCD and other LSDs with neurological compromise.

## 4. Material and Methods

### 4.1. Cell Culture

Cells were obtained from the NIGMS Human Genetic Cell Repository at the Coriell Institute for Medical Research. GM 05659 was used as a control (CTRL) cell line. Niemann–Pick C1 GM03123 fibroblasts with mutations in the NPC1 gene I1061T/P237S [73] were used. Human hepatocellular carcinoma Huh7 cells and human neuroblastoma cells SHSY-5Y were donated by Marco Arrese and Alejandra Alvarez, respectively (Pontificia Universidad Católica de Chile). HeLa cells stably expressing TFEB-GFP were donated by Andrea Ballabio (TIGEM, Naples, Italy). All protocols were approved by the Scientific Ethics Committee at Pontificia Universidad Católica de Chile (Number 150519019).

These cells were maintained in the Dulbecco’s modified eagle medium (DMEM) containing 10% fetal bovine serum from ThermoFisher Scientific ((Waltham, MA, USA), 100 µg/mL streptomycin, and 100 U/mL penicillin (Invitrogen, Waltham, MA, USA). All human fibroblasts and cells were cultured at 37 °C in a humidified atmosphere with 5% CO_2_.

### 4.2. Genistein and U18666A Treatments

Genistein was dissolved in DMSO at stock concentrations of 20 mM and stored at −20 °C. U18666A was dissolved in ethanol at stock concentrations of 1 mg/mL and stored at −20 °C. Huh7, SHSY-5Y, and HeLa TFEB-GFP cells were treated for a total of 24 h with and without U18 0.5 µg/mL, as well as in the presence or absence of genistein at a final concentration of 150 μM.

### 4.3. Reagents and Antibodies

Genistein (G6649), Filipin (F9765), and Chloroquine (C6628) were purchased from Sigma–Aldrich (St. Louis, MO, USA). Torin 1 (4247) was purchased from Tocris Bioscience (Minneapolis, MN, USA). U18666A (BML-S200) was purchased from Enzo Life Sciences Inc. (Farmingdale, NY, USA). The β-Hexosaminidase substrate 4-methylumbelliferyl *N*-acetyl-β-d-glucosaminide (M2133) was purchased from Sigma–Aldrich (St. Louis, MO, USA). Antibodies for rabbit anti-TFEB (4240) and rabbit anti-Histone H3 (9715) were purchased from Cell Signaling Technology (Danvers, Massachusetts, USA). Mouse anti-GAPDH (Human Glyceraldehyde-3-phosphate dehydrogenase) (sc-32233) was purchased from Santa Cruz Biotechnology (Dallas, TX, USA). Rabbit anti-LC3 (NB100-2220) was purchased from Novus Biologicals. Rabbit anti-p62 (56416) was purchased from Abcam (Cambridge, UK).

### 4.4. Nuclei-Cytoplasmic Fractions

Cells were seeded in 100 mm plastic dishes from Falcon (Tewksbury, MA, USA) and treated as indicated earlier, washed with PBS1X, and scraped gently. Nuclei–cytoplasmic fractions of cell lines and human fibroblasts were isolated as previously described [74].

In brief, samples were lysed in a buffer containing 10 mM Tris (pH 7.9), 140 mM KCI, 5 mM MgCl_2_, and 0.5 % NP-40 supplemented with fresh protease and phosphatase inhibitors. After 15 min, the lysate was centrifugated 500× *g* for 3 min at 4 °C. The supernatant was the cytoplasmic fraction. The cell pellet was washed 3 times with lysis buffer, was then incubated in 0.5 Triton X-100 buffer 0.5% SDS on ice for 20 min, and was gently sonicated on ice 3 times for 1 min. The supernatant was the nuclear fraction.

### 4.5. High Content Nuclear Translocation Assay

TFEB–GFP cells were seeded in 96-well plates, incubated for 24 h, and treated with 50, 100, and 150 µM of genistein alone, 0.5 µg/mL U18 and the same genistein concentrations plus U18 for 24 h. Torin 1 (0.3 µM) for 3 h served as a positive control for TFEB nuclear translocation. Cells were then washed and fixed in 4% paraformaldehyde/4% sucrose in PBS for 20 min.

Images were acquired on the Cytation 5 device and analyzed with the Gen 5 Image Prime v3.04.17 software. Twenty-five photos were taken per well in a 5 × 5 matrix with a magnification of 20× in 2 or 3 channels. The cells were segmented by a primary mask (nucleus), in which the intensity of the green signal corresponding to TFEB was measured under different conditions. Data are represented by mean ± SEM at different concentrations for each compound using the Prism software (GraphPad 6.0 software).

### 4.6. Immunoblot Analysis

Cells were lysed in RIPA buffer (25 mM Tris-HCl pH 7.6, 150 mM NaCl, 1% NP-40, 1% sodium deoxycholate, 0.1% SDS) supplemented with a protease inhibitor cocktail from Roche (Mannheim, Germany): Sodium Orthovanadate, phenylmethylsulfonyl fluoride (PMSF), and Pepstatin A. The homogenates were maintained on ice for 30 min and were then centrifuged at 10,000× *g* for 10 min. The supernatant was recovered, and protein concentration was determined with the Pierce BCA protein assay kit (23225) purchased from Fisher Scientific (Waltham, MA, USA). Proteins were resolved in SDS-PAGE, transferred to Nitrocellulose membranes from ThermoFisher Scientific (Waltham, MA, USA), blocked with 5% BSA in a PBS-T buffer (PBS containing 0.05% Tween-20), and probed with primary antibodies (1:1000) against TFEB, p62, LC3, GAPDH, and Histone H3 overnight. The reactions were followed by incubation with HRP labeled secondary antibodies (1:5000) at room temperature for 1 h and revealed using the enhanced chemiluminiscence ECL technique from (ThermoFisher Scientific (Waltham, MA, USA).

### 4.7. RNA Isolation

Three biological replicates of total RNA were extracted independently from 1 × 10^6^ human fibroblasts (WT and NPC patient) treated with genistein (150 µM) for 24 h using the TRIZOL Reagent from Invitrogen Waltham, MA, USA) and were purified using the RNeasy mini kit from Qiagen (Germantown, MD, USA) according to the manufacturer’s instructions. Total RNA was determined using a Qubit Fluorometric Quantitation System (Life Technologies), and purity (absorbance 260/280 nm) was established using a NanoQuant Spectrophotometer from Tecan Technologies (Männedorf, Switzerland), while integrity was confirmed by the RNA Integrity Number (RIN) using a 2200 TapeStation Instrument from Agilent Technologies (Santa Clara, CA, USA). Only high-quality samples (absorbance 260/280 nm ≥ 1.9 and RIN ≥ 8.5) were used for gene expression analyses.

### 4.8. cDNA Synthesis and Quantitative Polymerase Chain Reaction (qPCR)

Two μg of total RNA were used as templates for reverse transcription reactions to synthesize cDNA with a High-Capacity RNA to cDNA Kit from ThermoFisher Scientific (Waltham, MA, USA), according to standard procedures. cDNAs were diluted to 100 ng and used as templates for qPCR reactions that were carried out on a 96 real-time PCR System from Roche Life Science (Branchburg, NJ, USA) using the Brilliant II SYBR^®^ Green QPCR Master Mix from Agilent Technologies (Santa Clara, CA, USA). PCR conditions were 95 °C for 5 min followed by 94 °C for 15 s, 58–64 °C for 15 s, and 72 °C for 20 s for a total of 40 cycles. Melting curves (1 °C steps between 75–95 °C) ensured that a single product was amplified in each reaction. The method described by Pfaffl [75] and adapted by Talke and coworkers [76] was used to determine relative expression levels of the genes. Human Glyceraldehyde-3-phosphate dehydrogenase (GAPDH) was selected as the internal reference gene.

mRNA expression was analyzed for the following genes using appropriate primers: *TFEB*, Lysosomal-associated membrane protein 1 (*LAMP1*), Cathepsin B (*CTSB*), Cathepsin D (*CTSD*), Recombinant Human V-type proton ATPase subunit H (*HATP6V1H*), Tripeptidyl peptidase 1 (*TPP1*), Recombinant Human Autophagy-related protein 9B (*ATG9B*), Beclin-1 (*BECN1*), Mucolipin 1 protein (*MCOLN1*), Vacuolar protein sorting-associated protein 11 homolog (*VPS11*), and ATG18 protein homolog (*WIPI1*). Appendix A shows the complete list of primers used in this study.

### 4.9. Measurement β-Hexosaminidase Release

β-Hexosaminidase release was performed as described previously by Xu et al. with modifications [77]. In brief, NPC patient fibroblasts were cultured in 24-well plates at 30,000 cells/well in 0.3 mL of medium for 1 day at 37 °C. After being washed twice with assay buffer (DMEM with 2 mm D-mannose 6-phosphate sodium salts), the cells with 0.4 mL/well assay buffer were incubated for 40 min at 37 °C with 0.2 mL/well compound. Three aliquots of 25 μL assay buffer from each well in the 24-well plate were aliquoted into a 96-well black plate. The rest of the assay buffer in the 24-well plate was discarded, followed by the addition of 0.6 mL of 1% Triton X-100 in distilled H_2_O to lyse the cells. After incubation at 37 °C for 30 min, 5 μL/well cell lysate was added in triplicate to the 96-well plate with 20 μL of water, followed by 100 μL/well 1 mm β-Hexosaminidase substrate, 4-methylumbelliferyl *N*-acetyl-β-d-glucosaminide in a 0.4 m sodium acetate buffer at pH 4.4. The 96-well plate was then measured in the Synergy HT Biotek plate reader (excitation = 360 ± 40 nm, and emission = 460 ± 40 nm) after a 30 min incubation at 37 °C and the addition of 75 μL/well stop solution (0.5 m glycine and 0.5 m Na_2_CO_3_).

### 4.10. Filipin Staining

Filipin dye detects unesterified cholesterol in cells and was used as described previously [55]. Cells lines and NPC patient fibroblasts were fixed in 4% paraformaldehyde/4% sucrose in PBS for 30 min. Then, cells were washed with PBS and treated with 1.5 mg/mL glycine for 20 min. Finally, cells were treated with 25 μg/mL filipin for 30 min in the dark, washed with PBS, and covered with Fluoromount-G (SouthernBiotech, Birmingham, AL, USA). Images were captured with an Olympus BX51 microscope (Olympus, Tokyo, Japan) and analyzed with the ImageJ software.

## 5. Statistical Analysis

At least three biological replicates and three technical replicates were performed, and all values were represented as mean ± SEM. Comparisons between two groups were performed by the Student’s *t*-test, comparisons between several groups were tested by one-way analysis of variance (ANOVA), and the Tukey posttest was applied to perform a statistical comparison. Values of *p* < 0.05 were considered statistically significant. Data were processed using the GraphPad Prism 6 Software. PCR efficiencies were determined by linear regression analysis performed directly on the sample data using LinRegPCR [78].

## Figures and Tables

**Figure 1 ijms-22-04220-f001:**
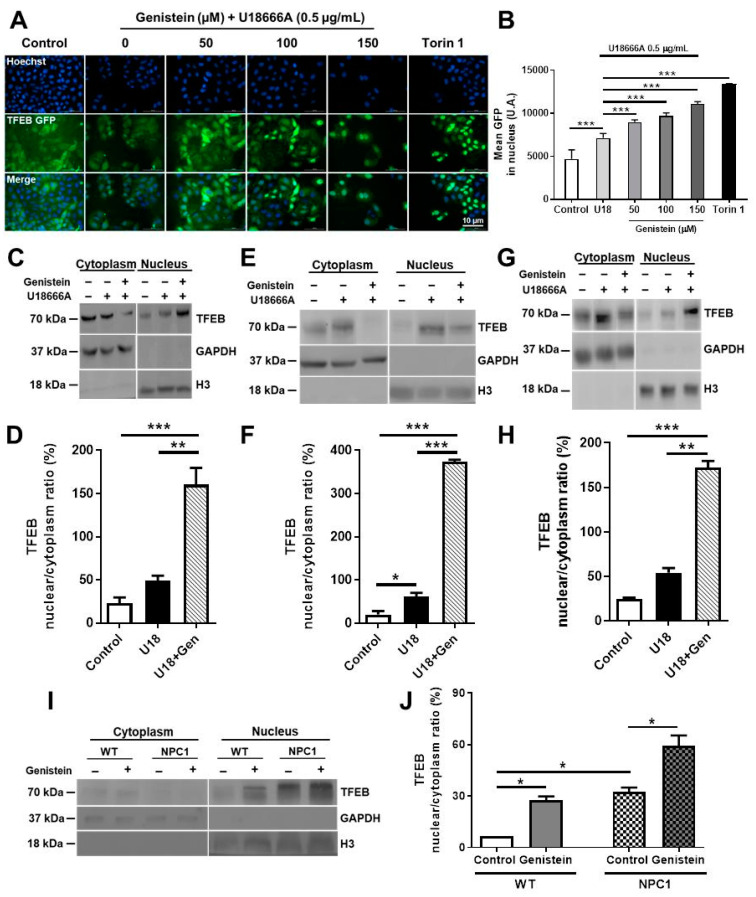
Genistein promotes transcription factor EB (TFEB) nuclear translocation to the nucleus in Niemann–Pick type C (NPC) cells. Representative images and quantification (**A**,**B**) of the TFEB-GFP translocation assay in HeLa cells stably expressing TFEB-GFP treated with 50, 100, and 150 µM of genistein, in control conditions or treated with U18666A (U18) 0.5 µg/mL for 24 h at the same time. As a positive control for TFEB nuclear translocation, we used 0.3 µM Torin 1 for 3 h. TFEB-GFP nuclear localization was analyzed using a high-content nuclear translocation assay in an epifluorescence automated microscope. Scale bar: 10 μm, GFP (green) and Hoechst (blue). For each condition, 3000–5000 cells were analyzed (five wells × 16 images). Representative immunoblot and quantification of endogenous TFEB levels in lysates from nuclear and cytoplasmatic fractions obtained after 24 h treatment with 150 µM of genistein, in control or 0.5 µg/mL of U18 treated HeLa cells stably expressing TFEB-GFP (**C**,**D**), Huh7 cells (**E**,**F**), SHSY-5Y cells (**G**,**H**), GM05659 WT, and the GM03123 NPC1 patient fibroblasts (**I**,**J**). Human Glyceraldehyde-3-phosphate dehydrogenase (GAPDH) and Histone H3 were used as cytoplasmatic and nuclear loading controls, respectively. * *p* ≤ 0.05, ** *p* ≤ 0.01, *** *p* ≤ 0.001 data represent mean ± SEM.

**Figure 2 ijms-22-04220-f002:**
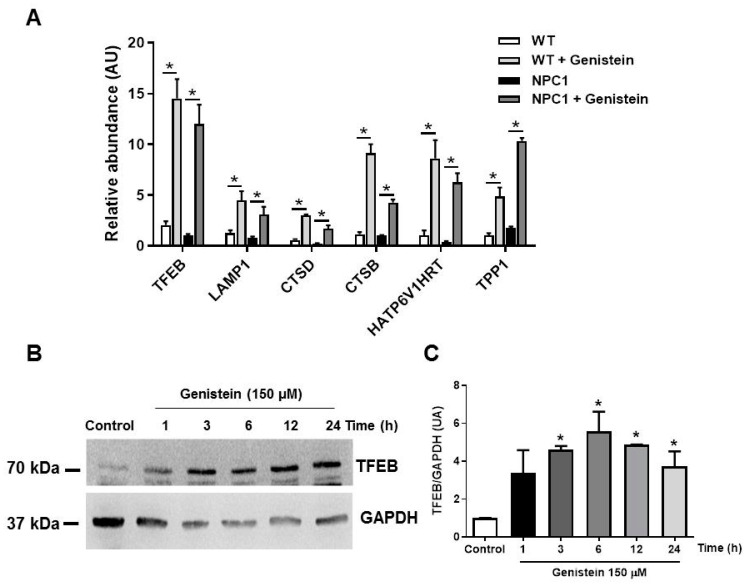
Genistein induces the mRNA expression of transcription factor EB (TFEB)-target genes and increases TFEB levels in NPC1 patient fibroblasts. (**A**) The graph shows qPCR analysis of mRNA levels of different TFEB target genes in GM05659 wild type (WT) fibroblast and GM03123 NPC1 patient fibroblasts after 24 h treatment with 150 µM of genistein. The bar graphs represent the relative fold induction of these genes normalized with *GAPDH* mRNA levels. Representative immunoblot (**B**) and quantification (**C**) of endogenous TFEB total levels in lysates from WT fibroblast and NPC1 patients fibroblast obtained after 1, 3, 6, 12, and 24 h treatment with 150 µM of genistein. GAPDH was used as a loading control. Data were presented as mean ± SEM of three independent samples (each with triplicates), * *p* ≤ 0.05, analyzed by ANOVA with Tukey posttest, *n* = 3 for each experiment (in triplicates for genes).

**Figure 3 ijms-22-04220-f003:**
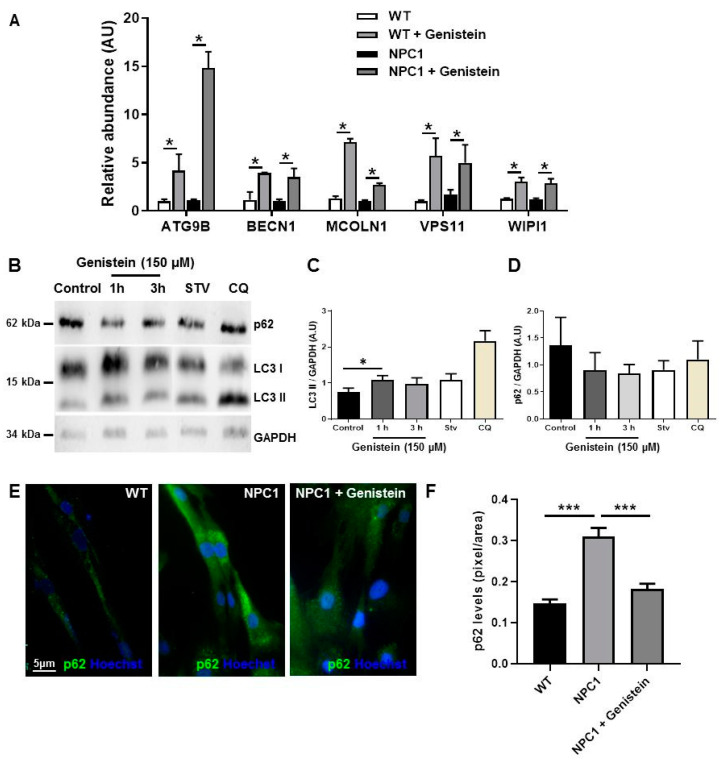
Genistein induces autophagy markers in NPC1 patient fibroblasts. The mRNA levels of the autophagy-related genes, *ATG9B*, *BECN1*, *MCOLN1, VSP11,* and *WIPI1,* were measured after 24 h of treatment with 150 µM of genistein by qPCR (**A**). Representative immunoblot (**B**) and quantification of LC3 (**C**) and p62 (**D**) levels in GM05659 wild type (WT) fibroblast and GM03123 NPC1 patient fibroblast obtained after 1 and 3 h treatment with 150 µM of genistein. As a positive control for autophagy inhibition and autophagy activation, we used chloroquine 20 µM and starvation media, respectively, for 3 h. GAPDH was used as loading control. Representative images (**E**) and quantification (**F**) of normalized relative fluorescence intensities for p62 on WT and NPC1 patient fibroblasts obtained after 3 h of treatment with 150 µM of genistein. The bar graphs show mean ± SEM, ** p* ≤ 0.05 and *** *p* ≤ 0.001 versus control analyzed by ANOVA with Tukey test, n = 3 for each experiment (in triplicates for genes).

**Figure 4 ijms-22-04220-f004:**
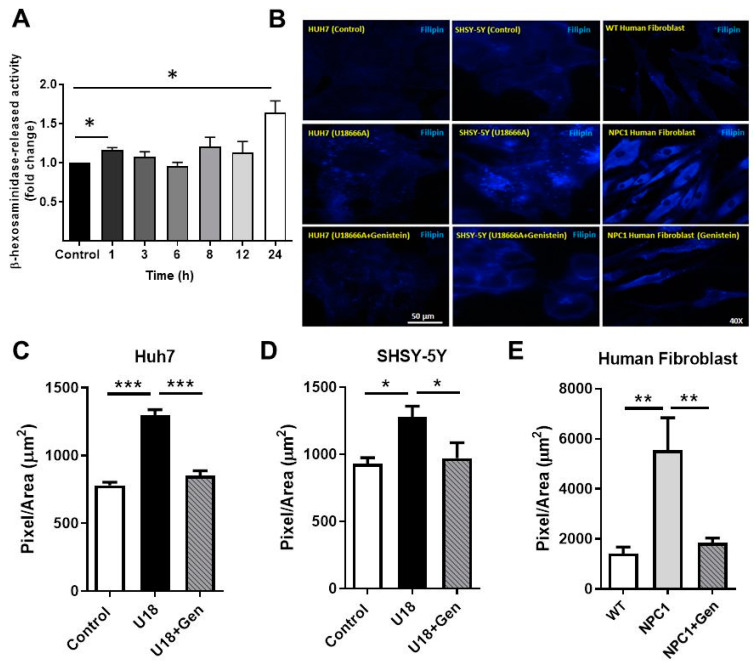
TFEB activation by genistein induces lysosomal exocytosis and reduces cholesterol accumulation in NPC models. (**A**) GM03123 NPC1 patient fibroblasts were cultured in 24-well plates at 30,000 cells/well and treated with 150 µM genistein for 1, 3, 6, 8, 12, and 24 h. The activity of the lysosomal enzyme β-Hexosaminidase was determined in both conditioned media and cell extracts. The bar graphs show the released enzyme activity measured as a percentage of total activity and expressed as fold change. Data are mean ± SEM from independent experiments, each in triplicate. Representative images and quantification of cholesterol accumulation by filipin staining in Huh7 (**B**,**C**) and SHSY-5Y (**B**–**D**) cells in control condition, treated with U18 (0.5 µg/mL) and U18 and genistein and WT (control) and NPC1 patient fibroblasts (**B**–**E**) treated with genistein 150 µM for 24 h. The filipin intensity quantification was made by the Image J program. The bar graphs show mean ± SEM filipin signal intensity normalized to controls, n = 5 images/condition. * *p* ≤ 0.05, ** *p* ≤ 0.01, *** *p* ≤ 0.001 versus control analyzed by ANOVA with Tukey posttest.

## Data Availability

Not applicable.

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
