# Peer review of "Genistein Activates Transcription Factor EB and Corrects Niemann–Pick C Phenotype"

_ijms, 2021, doi:10.3390/ijms22084220_

Round 1

Reviewer 1 Report

Manuscript ijms-1123227

' Genistein activates the Transcription Factor EB and corrects Niemann-Pick C phenotype' by  Argüello et al.

Comments to Authors:

This is a very interesting study on the effects of genistein on various models of Niemann-Pick type C disease. Although it is well designed and written, there are some points to be elucidated:

1) some English typos are present in the text

2) Authors do not indicate, if the project was accepted by a local bioethics committee, although the cell lines from patients were used in this study. It should be clearly explained if the project received such acceptance.

3) Authors performed experiments on only one cell line of each type and then conclude and discuss the results. This is rather generalization and I suggest to emphasize in Introduction and Results sections that experiments were done on 1 cell line of each NPC models.

4) There is only Fig.2 present in the manuscript. Other figures mentioned in the text are not available for review.

5) in the Discussion section it could be mentioned that disordered autophagy in NPC skin fibroblasts from 23 patients was also observed by Hetmanczyk-Sawicka et al. by means of changes in gene expression ('Changes in global gene expression indicate disordered autophagy, apoptosis and inflammatory processes and downregulation of cytoskeletal signalling and neuronal development in patients with Niemann-Pick C disease', DOI:  10.1007/s10048-019-00600-6

Author Response

Dear Reviewer #1

Thank you for reviewing our manuscript IJMS-1123227 entitled “Genistein Activates the Transcription Factor EB and Corrects Niemann-Pick C Phenotype.” In order to make our manuscript suitable for publication, we have considered the questions raised by you and we have modified the manuscript accordingly.

Please find below your comments and, in italic, our responses. To make it easier to identify the changes in the manuscript, we have highlighted them in yellow.

We thank you for the opportunity to improve our article and for considering its publication in this prestigious Journal.

Reviewer 2 Report

The paper by Arguello et al claims that genistein might help for the treatment of NPC diseases, the paper itself is not containing any new data regarding the genistein action which is known, furthermore,  clinical trials on different models of MPS did not show any activity of genistein. Moreover, authors make wrong statements such as: 

"It has been shown that genistein treatment can improve cognitive function, hyperactivity, and irritability and sleep disturbances in patients with Mucopolysaccharidosis (MPS) IV,  which is a LSD involving neurological compromise ".

It is well know that MPSIV does not have any CNS involvement. 

Genistein has been failing to all the promises done so far. Unless used in a serious clinical trial I would not publish anything making new hope on this system.

Author Response

Dear Reviewer #2

Thank you for reviewing our manuscript IJMS-1123227 entitled “Genistein Activates the Transcription Factor EB and Corrects Niemann-Pick C Phenotype.” In order to make our manuscript suitable for publication, we have considered the questions raised by you and we have modified the manuscript accordingly.

Please find below your comments and, in italic, our responses. To make it easier to identify the changes in the manuscript, we have highlighted them in yellow.

We thank you for the opportunity to improve our article and for considering its publication in this prestigious Journal.
